# Systemic Dysfunction of Osteoblast Differentiation in Adipose-Derived Stem Cells from Patients with Multiple Myeloma

**DOI:** 10.3390/cells8050441

**Published:** 2019-05-10

**Authors:** Véronique Béréziat, Christelle Mazurier, Martine Auclair, Nathalie Ferrand, Séverine Jolly, Tiffany Marie, Ladan Kobari, Indira Toillon, François Delhommeau, Bruno Fève, Annette K. Larsen, Michèle Sabbah, Laurent Garderet

**Affiliations:** 1INSERM, UMR_S 938, Centre de Recherche Saint-Antoine-Team Genetic and Acquired Lipodystrophies, Institut Hospitalo-Universitaire de Cardiométabolisme et Nutrition (ICAN), Sorbonne Université, F-75012 Paris, France; veronique.bereziat@inserm.fr (V.B.); martine.auclair@inserm.fr (M.A.); indira.toillon@inserm.fr (I.T.); 2INSERM, UMR_S 938, Centre de Recherche Saint-Antoine-Team Proliferation and Differentiation of Stem Cells, Institut Universitaire de Cancérologie, Sorbonne Université, F-75012 Paris, France; cmazurier@hotmail.com (C.M.); severine_jolly@yahoo.fr (S.J.); tiffany.marie@orange.fr (T.M.); 3EFS Ile de France, Unité d’Ingénierie et de Thérapie Cellulaire, F-94017 Créteil, France; 4INSERM, CNRS, UMR_S 938, Centre de Recherche Saint-Antoine- Team Cancer Biology and Therapeutics, Institut Universitaire de Cancérologie, Sorbonne Université, F-75012 Paris, France; nathalie.ferrand@inserm.fr (N.F.); annette.larsen@mfex.com (A.K.L); michele.sabbah@inserm.fr (M.S.); 5INSERM, UMR_S 938, Centre de Recherche Saint-Antoine-Team Proliferation and Differentiation of Stem Cells, Institut Universitaire de Cancérologie, Sorbonne Université, F-75012 Paris, France; ladan.kobari@sorbonne-universite.fr (L.K.); francois.delhommeau@aphp.fr (F.D.); 6INSERM, UMR_S 938, Centre de Recherche Saint-Antoine-Team Genetic and Acquired Lipodystrophies, Institut Hospitalo-Universitaire de Cardiométabolisme et Nutrition (ICAN), Assistance Publique-Hôpitaux de Paris, Hôpital Saint-Antoine, service d’Endocrinologie, Sorbonne Université, F-75012 Paris, France; bruno.feve@inserm.fr; 7INSERM, UMR_S 938, Centre de Recherche Saint-Antoine-Team Proliferation and Differentiation of Stem Cells, Assistance Publique-Hôpitaux de Paris, Hôpital Saint Antoine, Département d’Hématologie et de Thérapie Cellulaire, Sorbonne Université, F-75012 Paris, France

**Keywords:** multiple myeloma, bone disease, osteogenesis, adipogenesis, adipose-derived stem cells, bone marrow, senescence, Dickkopf-related protein 1, systemic disease

## Abstract

Multiple myeloma is characterized by bone lesions linked to increased osteoclast and decreased osteoblast activities. In particular, the osteoblast differentiation of bone marrow-derived stem cells (MSC) is impaired. Among the potential therapeutic tools for counteracting bone lesions, adipose-derived stem cells (ASC) could represent an appealing source for regenerative medicine due to their similar characteristics with MSC. Our study is among the first giving detailed insights into the osteoblastogenic capacities of ASC isolated by fat aspiration from myeloma patients (MM-ASC) compared to healthy subjects (HD-ASC). We showed that MM-ASC and HD-ASC exhibited comparable morphology, proliferative capacity, and immunophenotype. Unexpectedly, although normal in adipocyte differentiation, MM-ASC present a defective osteoblast differentiation, as indicated by less calcium deposition, decreased alkaline phosphatase activity, and downregulation of RUNX2 and osteocalcin. Furthermore, these ASC-derived osteoblasts displayed enhanced senescence, as shown by an increased β-galactosidase activity and cell cycle inhibitors expression (p16^INK4A^, p21^WAF1/CIP1^.), associated with a markedly increased expression of DKK1, a major inhibitor of osteoblastogenesis in multiple myeloma. Interestingly, inhibition of DKK1 attenuated senescence and rescued osteoblast differentiation, highlighting its key role. Our findings show, for the first time, that multiple myeloma is a systemic disease and suggest that ASC from patients would be unsuitable for tissue engineering designed to treat myeloma-associated bone disease.

## 1. Introduction

Multiple myeloma (MM) is the most frequent malignant hematological disease, after non-Hodgkin’s lymphoma, and represents 13% of all malignant hematological disorders, corresponding to approximately 115,000 new cases and 80,000 deaths yearly, worldwide [1]. MM is associated with severe malignancy of the bone marrow, incurable in most cases. MM is characterized by clonal proliferation of plasma cells in the bone marrow and the secretion of monoclonal immunoglobulins. In the majority of cases, the symptoms are dominated by lytic bone lesions and may also be accompanied by anemia, renal failure, and/or metabolic manifestations, such as hypercalcemia [2]. 

MM-associated bone disease (MMBD) is the signature of MM and results from persistent uncoupling of the normal bone remodeling process, including increased osteoclast and attenuated osteoblast activities [3,4,5]. The bone lesions in patients with MM are irreversible, deteriorate during disease progression, and represent a major component of the morbidity. MMBD is currently treated with bisphosphonates and, to some extent, using targeted therapies, such as lenalidomide and bortezomib. Bisphosphonates are used to prevent and slow down lytic bone lesion [6,7]. They also correct hypercalcemia and reduce bone pain with less need for pain-killers. Nevertheless, no really effective treatment exists to restore bone integrity after osteolytic lesion. Since MMBD is responsible for the most deleterious complications of MM, it is essential to develop new therapeutic strategies to combat this destructive process. 

We and others have reported a role of bone marrow-derived MSC in the pathophysiology of MM. Indeed, bone marrow-derived MSC can stimulate MM cell growth, invasion, angiogenesis, and drug resistance. Furthermore, we found that MSC derived from the bone marrow of MM patients have a much lower proliferative capacity than cells from healthy individuals and overproduce Dickkopf-related protein 1 (DKK1), a Wnt-pathway antagonist [8,9,10]. DKK1 acts as a major inhibitor of osteoblastogenesis in MM by sequestering LRP5/6 (low density lipoprotein receptor-related protein), leading to downregulation of the osteoblast transcription factor RUNX2 [11]. Therefore, contributing significantly to the oncogenesis of MM, bone marrow-derived MSC from MM patients are unsuitable for autologous bone reconstruction [12].

In contrast, MSC originated from adipose tissue might be suitable for tissue engineering, particularly as these regions lie distant from the pathological medullary microenvironment. Adipose tissue represents an appealing source for regenerative medicine since it is abundant, ubiquitous, and relatively simple to access [13]. Adipose-derived stem cells (ASC) display similar stromal cell properties as their bone marrow-derived MSC counterparts, i.e., (i) self-renewal capacity and (ii) multipotency for the adipocyte, chondrocyte, or osteoblast lineages [14,15]. Similar to MSC, ASC maintain their pluripotency during expansion under standard culture conditions and phenotypically retain markers in common with MSC, such as CD90, CD73, and CD105, but not CD45. However, ASC may be distinguished from bone marrow-derived MSC through their expression of CD36 in the absence of CD106 [16,17].

In order to test the potential use of ASC as therapeutic tool for counteracting the bone lesions observed in patients, we analyzed the osteoblastogenic potential of ASC in both healthy donor and MM patients. We report here, for the first time that, surprisingly, ASC derived from patients with MM exhibit defective osteoblast differentiation while retaining their adipogenic capacity. In particular, ASC from these patients displayed increased senescence and expressed high levels of DKK1. These findings indicate that ASC from MM patients would be unsuitable for use as osteoblast precursors for autologous bone repair cell therapy.

## 2. Materials and Methods

### 2.1. Patients 

The characteristics of the patients are listed in Appendix A. The median age was 68 years (range 50–84), which is the usual age for the occurrence of myeloma, and 7 in 10 patients had bone lesions. The median medullary plasmocytosis was 27% (range 6–80).

### 2.2. Cell Sample Collection

Normal adipose tissues were obtained from healthy donors undergoing liposuction procedures (Clinique de l’Alma, Paris 7, France). Otherwise, human abdominal subcutaneous adipose tissue samples used for adipose stem cells (ASC) isolation were obtained from 6 healthy donors (three women and three men). The body mass index was 25.6 ± 0.6 kg/m^2^ with a mean age of 36.9 ± 6.05 years.

Myeloma adipose tissues were obtained from 11 newly diagnosed MM patients by abdominal fat aspiration (AP-HP Saint-Antoine Hospital, Paris, France). Myeloma bone marrows were collected by sternal puncture. Both normal and myeloma abdominal fat and bone marrow cell samples were obtained with the free and informed consent of the donors/patients, in accordance with the ethical standards of the local ethical committee and the Declaration of Helsinki (1964). The study was approved by the French regulatory authorities (CPP Ile de France V, n° 14964).

### 2.3. Mesenchymal Stromal Cell Isolation and Culture

ASC from healthy and MM donors were isolated using a collagenase digestion, as previously described [18,19], and seeded in αMEM media supplemented with 5% human platelet lysate (hPL) (MacoPharma Biotech, Tourcoing, France) and 2 IU/mL heparin (Panpharma, Luitré, France). At confluence, adherent cells were harvested (0.05% trypsin-EDTA, Gibco), counted by trypan blue exclusion, and reseeded at 10^3^ cells/cm^2^ for expansion. The culture medium was changed twice a week. Bone marrow-derived MSC from myeloma donors were seeded in αMEM supplemented with 5% hPL, 0.5% ciprofloxacine (Bayer Pharma, Leverkusen, Germany), and 2 IU/mL heparin. All experiments were performed with cells isolated from at least four different healthy or MM donors.

### 2.4. Mesenchymal Stromal Cell Immunophenotyping

ASC phenotypes were analyzed at passage 2 (P2) by flow cytometry using the following antibodies: FITC-conjugated anti-CD36, anti-CD38, anti-CD45 and anti-CD14, PE-conjugated anti-CD73, anti-CD90, anti-CD105, anti-CD106 and anti-CD34, PB-conjugated anti-HLA-DR, APC-conjugated anti-CD19, and PC5-conjugated anti-CD11b clone. The corresponding irrelevant isotypes were used in accordance with the manufacturer’s instructions (BD Sciences, Franklin Lakes, NY, USA). Briefly, once detached from their support, the cells were washed in PBS 1X containing 4% fetal bovine serum (FBS) and the cell concentration was adjusted to 1 × 10^6^ cells/mL. Aliquots of 1 × 10^5^ cells were incubated for 15 min, at room temperature, with the appropriate quantities of antibodies according to the manufacturer’s guidelines and then washed and resuspended in 100 µL of PBS containing 4% FBS for flow cytometric analysis.

### 2.5. Cell Differentiation

For differentiation assays, cells at passage 2 were used. Differentiation of ASC was induced by addition of a proadipogenic medium for 4 days (DMEM with 10% FBS, 1 µM dexamethasone, 500 µM isobutylmethylxanthine, 1 µM insulin, and 1 µM rosiglitazone (all from Sigma-Aldrich, Saint-Louis, MO, USA) and then maintained in DMEM (1 µM insulin and 1 µM rosiglitazone). After 14 days, adipogenic differentiation was evaluated by measuring the lipid accumulation using Oil-red-O (Sigma-Aldrich), as described previously [19,20]. Cells were differentiated into osteoblasts, as previously described [21]. After 15 days, osteoblast differentiation was evaluated by measuring alkaline phosphatase activity according to the manufacturer’s protocol (Sigma-Aldrich) and calcium deposition using Alizarin Red staining (Sigma-Aldrich).

### 2.6. Cell Senescence

A positive blue staining for β-galactosidase was used as a biomarker for cell senescence [22]. To detect senescence-associated β-galactosidase activity, the cells were incubated in a buffer solution containing X-Gal (Sigma-Aldrich), as described elsewhere [23].

### 2.7. Western Blotting

Cellular extracts were prepared, as previously described [23], and immunoblotting was done via standard procedures, according to the manufacturer’s instructions. Adipocyte differentiation markers were identified using antibodies against CCAAT/enhancer binding protein α (C/EBPα, Santa Cruz Biotechnology, Dallas, TX, USA) and peroxisome proliferator-activated receptor γ (PPARγ, Santa Cruz Biotechnology). Osteoblast differentiation markers were detected using antibodies against osteocalcin (Santa Cruz Biotechnology), Runt-related transcription factor 2 (RUNX2, R&D Systems, Minneapolis, MN, USA), and Dickkopf-related protein 1 (DKK1, R&D Systems). The expression of senescence-associated cell cycle inhibitors was determined using antibodies against p16^INK4A^ and p21^WAF1/CIP1^ (BD-Pharmingen, BD Biosciences, San Jose, CA, USA). Tubulin (Sigma Aldrich) was used as an internal control for protein loading. The antibodies were detected with a chemiluminescence detection kit (Amersham Biosciences GE Healthcare Europe, Velizy Villacoublay, France). Western blot quantification was performed in triplicate using Fiji software (Open source) and results were normalized to the tubulin protein levels.

### 2.8. Statistical Analyses

Quantitative results were expressed as the mean ± SEM. Statistical analyses were performed using the non-parametric Mann–Whitney test and *p* values of less than 0.05 were considered to be statistically significant. 

## 3. Results

### 3.1. ASC from Healthy Donors and MM Patients are Comparable with Respect to Morphology, Phenotype and Proliferative Capacity

Firstly, the ASC populations were characterized according to the criteria of the International Society for Cellular Therapy (ISCT) [14]. ASC from both healthy donors (HD-ASCs) (Figure 1A, left panels) and MM patients (MM-ASC) (Figure 1A, right panels) adhered to plastic culture plates when maintained under standard culture conditions and displayed a typical fibroblast-like morphology under the light microscopy (Figure 1A). No significant morphological modifications were observed during cell culture, whatever the passage or the source of the cells. 

We next evaluated the cumulative proliferative capacity between passages 1 and 3 by counting the number of viable cells. As shown in Figure 1B, MM-ASC and HD-ASC exhibited comparable in vitro growth, especially at early passages (P1: 26.4 ± 3.4 vs. 29.4 ± 3.2-fold expansion; P1-P2: 543.6 ± 91.4 vs. 648.3 ± 126.2-fold expansion; P1-P2-P3: 7596.5 ± 1674.9 vs. 12975.8 ± 3308-fold expansion, MM-ASC vs. HD-ASC, respectively). Although statistical significance was not reached, the mean doubling time was nevertheless clearly longer for MM-ASC (P1: 55.7 ± 5.1 h; P2: 85.0 ± 6.5 h; P3: 116.3 ± 14.2 h) as compared to HD-ASC (P1: 47.1 ± 6.6 h; P2: 60.2 ± 9.2 h; P3: 86.8 ± 24.2 h), particularly at passages 2 and 3.

The phenotypes of the two types of ASC were then determined at the second passage by flow cytometry using a panel of surface markers selected according to the definition of the classical mesenchymal immune phenotype. Careful examination was paid to avoid cell doublets and the gating strategy was based on singlets after elimination of dead cells and cellular debris. ASC were positive for CD90, CD105, and CD73 (expression >97%), and negative for the hematopoietic markers CD45, CD14, HLA-DR, CD11b, and CD19 (expression <0.5%), without any detectable differences between cells from healthy donors or MM patients, in terms of either the percentage of positive cells or the mean fluorescence intensity (Table 1 and Figure 1C). Both HD-ASC and MM-ASC expressed very low levels of CD34 (Table 1), while CD38 expression was negative in both cases, indicating that the cell cultures contained no myeloma cells (data not shown). We also analyzed the expression of CD106 and CD36, which are thought to be differentially expressed by MSC derived from bone marrow, as compared to adipose tissue. No statistically significant differences were observed for CD106 between HD-ASC and MM-ASC. In contrast, MM-ASC cultures contained significantly greater numbers of CD36-positive cells, displaying a higher mean fluorescence intensity than the cells in HD-ASC cultures (Figure 1C).

### 3.2. ASC from MM Patients are Capable of Normal Adipocyte Differentiation

After reaching confluence at the second passage, ASC were induced to differentiate into adipocytes for 14 days. The adipogenic capacity of the cells was evaluated by measuring their lipid accumulation using Oil-red-O staining and their expression of the adipocyte-specific transcription factors PPARγ and C/EBPα. As shown in Figure 2, HD-ASC and MM-ASC exhibited a comparable significant increase in lipid accumulation between D0 and D14 of adipocyte differentiation. Similarly, the expression of PPARγ and C/EBPα and the secretion of adiponectin increased throughout the differentiation process, with no significant differences between HD-ASC and MM-ASC (Figure 2B and data not shown). Interestingly, the adipogenic capacities of MM-ASC were the same regardless of the status of the bone lesions (Appendix A).

### 3.3. ASC from MM Patients Display Defective Osteoblast Differentiation

Next, we investigated the capacity of the cells to differentiate into osteoblasts. Unexpectedly, as compared to HD-ASC, MM-ASC displayed strongly reduced calcium deposition, as assessed by Alizarin Red staining, as well as low alkaline phosphatase activity (Figure 3A). In addition, we observed no increased in RUNX2 or osteocalcin expression in MM-ASC cultures, unlike in HD-ASCs controls (Figure 3B). Furthermore, strong expression of Dickkopf-related protein 1, a major inhibitor of osteoblastogenesis, was observed in MM-ASC cultures throughout the entire differentiation process (Figure 3B), while, as expected, DKK1 was virtually undetected in HD-ASC. Importantly, these alterations were similar regardless of the bone lesions observed (Appendix A) nor the age of MM patients (data not shown), suggesting that the defective osteoblast differentiation of MM-ASC was an early dysfunction that is not age-related. Altogether, these results clearly indicated that MM-ASC have a reduced capacity to differentiate into osteoblasts.

### 3.4. ASC from MM Patients Exhibit a Senescent Phenotype

We then analyzed cellular senescence. HD-ASC cultures contained only low levels of senescent cells, as shown by their low SA-β-Galactosidase activity. In contrast, the number of senescent cells was 3-fold higher in MM-ASC cultures (Figure 4A). Accordingly, we observed increased expression of the cell cycle inhibitors p21^CIP1/WAF1^ and p16^INK4^, which are generally associated with senescence, in MM-ASC cultures (Figure 4B). These results were once again independent of the bone lesions status (Appendix A) and the age of the donor or patient (data not shown). Thus, our results showed that MM-ASC-derived osteoblasts display abnormally high levels of cellular senescence markers.

### 3.5. ASC and MSC from MM Patients Present Comparable Defects

Finally, we investigated whether the alterations observed in MM-ASC were similar to those found in MM-MSC isolated from the bone marrow. A comparative analysis was carried out using MSC and ASC obtained by bone marrow aspiration or liposuction from the same MM patients. Adipocyte differentiation was functional in both MM-ASC and MM-MSC, as evidenced by Oil-red-O staining and the quantification of the adipogenic transcription factors PPARɣ and C/EBPα (Figure 5A and Appendix A). However, osteoblastic differentiation was defective in both ASC and MSC from these patients, as shown by their decreased calcium deposition, low alkaline phosphatase activity, and borderline expression of the osteoblastic transcription factors RUNX2 and osteocalcin (Figure 5B and Appendix A). Interestingly, both MM-ASC and MM-MSC displayed enhanced SA-β-Galactosidase activity during osteoblastic differentiation and increased expression of the cell cycle inhibitors p21^CIP1/WAF1^ and p16^INK4^ (Figure 5C and Appendix A). These findings indicated that MM-MSC and MM-ASC derived from the same donor were comparable with regard to adipocyte and osteoblast differentiation.

### 3.6. DKK1 Inhibition Rescues RUNX2 Expression

To explore a potential causal role of secreted DKK1 in the osteoblastic differentiation alterations of MM-ASC, we performed rescue experiments by adding specific DKK1-neutralizing antibodies to the MM-ASC cultures. Notably, inhibition of DKK1 led to increased RUNX2 expression, accompanied by reduced expression of p21^CIP1/WAF1^ and p16^INK4^ (Figure 6A and Appendix A). Conversely, HD-ASC incubated with recombinant DKK1 exhibited a strong decrease in RUNX2 expression associated with increased expression of p21^CIP1/WAF1^ and p16^INK4^ (Figure 6B and Appendix A). Interestingly, an effect similar to that of recombinant DKK1 was observed when HD-ASC cells were incubated in the presence of conditioned medium derived from MM-ASC cultures. Thus, the conditioned medium suppressed RUNX2 expression and promoted p21^CIP1/WAF1^ and p16^INK4^ expression, an effect which was prevented by addition of anti-DKK1 antibodies to the conditioned medium (Figure 6B and Appendix A). These findings showed, for the first time, that MM patients express high levels of DKK1 protein in extra-hematopoietic tissue, which is suggestive of a systemic disorder. Our results further suggest that the DKK1-associated osteoblastic differentiation defects are mediated, at least in part, by enhanced cellular senescence.

## 4. Discussion

Multiple myeloma is characterized by deleterious bone lesions. Specifically, the malignant plasmocytes interact with the bone marrow stromal cells, leading to bone lesions through osteoclast activation and impairment of osteoblast-mediated bone remodeling [24]. We show here, for the first time, that stem cells at sites distant from the bone marrow, namely in adipose tissue, display similarly defective osteoblastic differentiation, although they are capable of normal adipocyte differentiation.

Although adipose-derived stromal cells from healthy subjects (HD-ASC) and myeloma patients (MM-ASC) were comparable in many aspects, including their phenotype and population doubling time at early passages, distinct differences appeared once the cells had been induced to undergo osteoblastic differentiation. In particular, calcium deposition and alkaline phosphatase activity were strikingly decreased in cells derived from the adipose tissue of MM patients, as compared to that of healthy donors. Furthermore, the transcription factor RUNX2 and the secreted factor osteocalcin were markedly under-expressed in cells derived from myeloma patients. Interestingly, the abnormal osteoblastic differentiation was associated with MM-ASC senescence, as shown by the increased senescence-associated β-galactosidase activity of the cells and their expression of the cell cycle inhibitors p21^CIP1/WAF1^ and p16^INK4^. Interestingly, patients with no bone lesions, at least at diagnosis and with the skeletal exams performed, still have a similar defect of osteoblast differentiation together with an increased senescence, as compared to patients with bone lesions. These data suggest that dysfunction of ASC is observed in MM patients even without bone disease.

The decreased osteogenic potential of MSC from the bone marrow of myeloma patients has been described by different groups and appears to be associated with both increased expression of inflammatory cytokines [25] and inhibition of the canonical Wnt/beta-Catenin pathway required for osteoblastic differentiation [26]. However, we did not expect to find a defect in MSC function at a site distant from the bone marrow medullary environment. This suggests that myeloma may be a systemic disease, rather than being restricted to the bone marrow. Such a hypothesis is supported by the fact that, in the same patient, similar alterations were observed in ASC, derived from the adipose tissue, or in MSC, derived from bone marrow.

Various mechanisms might explain the generalized influence of the modified bone marrow microenvironment. The simplest would be that MSC migrate from the bone marrow through the blood to the adipose tissue. However, this hypothesis is unlikely since MM-ASC are characterized by high CD36 and low CD106 expression, a phenotype which clearly distinguishes them from bone marrow-derived MSC [16]. Moreover, positivity for CD36 has been shown to be associated with strong adipogenic capacity [27]. Alternatively, small molecules like microRNA (miRNA) and/or MSC-derived extracellular vesicles, including exosomes and microvesicles originating from MSC or malignant plasmocytes within the bone marrow microenvironment, might target distant tissues such as the adipose tissue [28,29,30,31,32]. Interestingly, it has been suggested that exosomes derived from acute myeloid leukemia (AML) cells are able to induce expression of the Wnt-signaling inhibitor DKK1 in bone marrow stromal cells, thereby promoting osteoblast loss [33].

DKK1 has previously been implicated in the osteolytic phenotype of multiple myeloma [26,34] and elevated serum DKK-1 levels correlate with the presence of focal bone lesions as detected by MRI [26]. Our study revealed very high levels of DKK1 in conditioned medium from MM-ASC cultures, as compared to HD-ASC cultures. Furthermore, we found that the inhibition of DKK1 in MM-ASC leads to the re-expression of RUNX2, together with downregulation of the cell cycle inhibitors p21^CIP1/WAF1^ and p16^INK4^. Conversely, the addition of DKK1 protein or conditioned medium from MM-ASC cultures downregulated RUNX2 and increased the expression of p21^CIP1/WAF1^ and p16^INK4^. To confirm its key role, it would be very interesting to analyze DKK1 in situ expression in adipose tissue samples from MM patients. It is noteworthy that DKK1 is able to induce the proliferation of MSC and to switch their differentiation pathway from osteogenesis to adipogenesis [34,35]. As a result, patients recently diagnosed with MM have more abdominal fat and a higher fat metabolic activity than patients with monoclonal gammopathy of unknown significance, suggesting a switch to adipogenesis [36].

CD36 expression has recently been reported to increase the production of many factors associated with the senescence-related secretory phenotype through activation of the canonical Src-p38-NF-κΒ signaling pathway, which results in the onset of a fully senescent state [37]. This suggests that overexpression of DKK1 and CD36 might cooperate through different signaling pathways to inhibit osteoblastic differentiation in MM-ASC. Besides DKK1, we speculate that another soluble Wnt antagonist, secreted Frizzled-related protein 2 (sFRP-2), could also be involved in the inhibition of the osteoblastic differentiation. Thus, in vitro suppression of the mineralization and alkaline phosphatase activity of osteoblasts has been correlated with sFRP-2 secretion in myeloma cell lines and primary myeloma cells from patients with advanced bone lesions [38]. Altogether, these observations suggest that the osteoblastic defects of MM are at least partly associated with the capacity of MM cells to secrete inhibitors of the Wnt-signaling pathway.

Interestingly, enhanced expression of DKK1 has previously been implicated in the perturbation of normal cell differentiation in a different pathology, namely the dysfunction of hair follicles associated with male baldness. Hair follicles usually undergo a cycle of growth (anagen), regression (catagen), and rest (telogen). Higher production of DKK1 promotes a premature transition from anagen to catagen, probably by increasing expression of the proapoptotic protein Bax, thereby inducing apoptosis in the keratinocytes of the outer root sheath. Notably, injection of recombinant human DKK1 (rhDKK1) into the hypodermis of mice during anagen caused the premature onset of catagen, whereas a neutralizing anti-DKK1 antibody delayed catagen progression in mice [39]. DKK1 secretion has also been associated with increased cellular senescence in other pathologies, such as human reflux esophagitis [40] and in melanocytes, where it promotes the senescent phenotype [41]. The development of fracture non-union, with a localized reduced capacity of stem cells to undergo osteogenesis, is also associated with increased cell senescence and DKK1 secretion [42]. In myeloma, DKK1 could be at the origin of ASC senescence that we observed.

We acknowledge that our study has some limitations which have to be pointed out. The number of patients is relatively small and larger series are needed to confirm our results. However, we analyzed samples from both women and men and all samples were withdrawn by fat aspiration at similar locations. AT is a heterogeneous tissue composed of two cell fractions, the adipocyte and the stroma-vascular fraction (SVF), that mainly includes fibroblasts, endothelial cells, immune cells, and precursor cells, such as ASC. All these cellular components are useful to ensure the metabolic functions of AT. Nevertheless, ASC are multipotent cells that represent an adequate model for studying osteoblastogenesis. Primary culture is a more physiological model then cell lines, especially when studying pathophysiological mechanisms, but it requires precautions of use. Specifically, extraction protocols, culture conditions, number of passages, and donor are all variables to consider. To that aim, all the experiments were performed at passage two. The mean age of healthy donors and patients is quite different. It is quite difficult to obtain adipose tissue from younger patients as the mean age at diagnosis of myeloma is 70 years. However, when performed with ASC from younger patients, experiments showed similar data, suggesting that the alterations of ASC that we observed in patients were not age-related.

## 5. Conclusions

In summary, our results indicate that adipose-derived stem cells from myeloma patients are not suitable for osteoblastic differentiation and bone reconstruction. Importantly, these findings imply that myeloma should be regarded as a systemic disease. The observed defect in osteoblastic differentiation at sites distant from the bone marrow would seem to be at least partly related to overexpression of DKK1, which clearly points to a causal role of Wnt-signaling inhibitors in the osteoblastic defects of MM, but also cellular senescence process.

## Figures and Tables

**Figure 1 cells-08-00441-f001:**
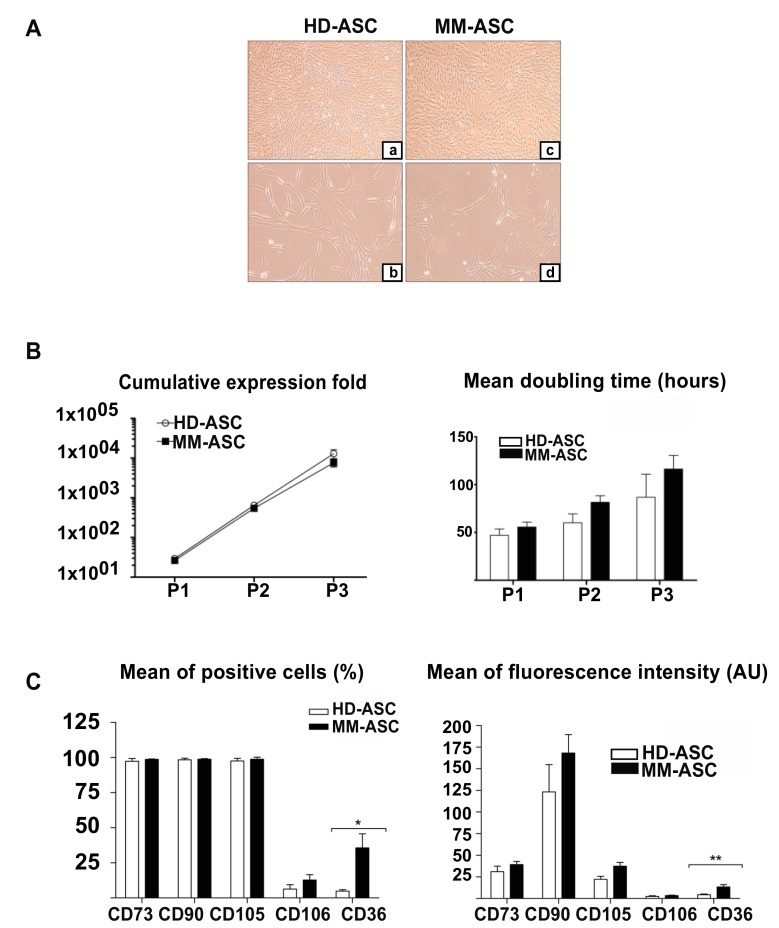
MM-ASC have normal morphology, proliferation capacity, and immunophenotype. (**A**) Morphology of the different stem cell populations. HD-ASC (**a** and **b**) and MM-ASC (**c** and **d**) were visualized at 2× (**a** and **c**) or 10× (**b** and **d**) magnification, using standard light microscopy. (**B**) Proliferative capacity of HD-ASC (*n* = 6) and MM-ASC (*n* = 11). Left: Mean cumulative expansion rate between P1 and P3. The number of viable cells (Trypan blue staining) was determined at the end of each passage (at confluence) and the cumulative expansion was calculated as the ratio of the total number of cells collected at the end of the passage to the total number of cells plated. Right: Mean doubling time calculated for each passage as follows: Doubling time = (ΔT × ln2)/(ln (Nn) – ln (N0)), where Nn is the number of cells at confluence and N0 is the number of cells seeded. Results are expressed as the mean ± SEM; * *p* < 0.05, using an unpaired t-test with Welch’s correction. (**C**) Immunophenotypes of HD-ASC (*n* = 6) and MM-ASCs (*n* = 11) at passage 2. The percentage of positive cells (%) (left) and the mean fluorescence intensity in arbitrary units (AU) (right) are indicated for each hematopoietic marker. Results are expressed as the mean ± SEM, * *p* < 0.05, MM-ASC vs. HD-ASC using unpaired *t*-test with Welch’s correction. AU: Arbitrary units.

**Figure 2 cells-08-00441-f002:**
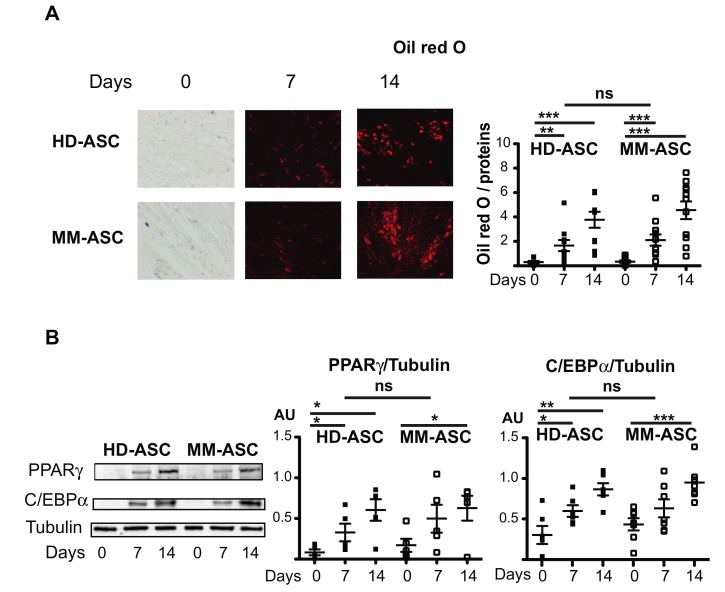
Adipocyte differentiation is functional in MM patients. HD- and MM-ASC were differentiated into adipocytes for 7 or 14 days. (**A**) The cells were stained with Oil-red-O to visualize lipid droplets after 7 or 14 days of differentiation and representative micrographs and scans are shown (left). Staining was quantified at 520 nm and normalized to the protein content (right). (**B**) Whole cell lysates were extracted on day 0, 7, or 14 of differentiation and analyzed by immunoblotting. Representative immunoblots of PPARγ, C/EBPα, and tubulin (loading control) are shown in the left panel and Western blot quantifications in the two right panels. AU: arbitrary units. * *p* < 0.05, ** *p* < 0.01, *** *p* < 0.001, vs. day 0. NS, not significant. HD-ASC vs. MM-MSC.

**Figure 3 cells-08-00441-f003:**
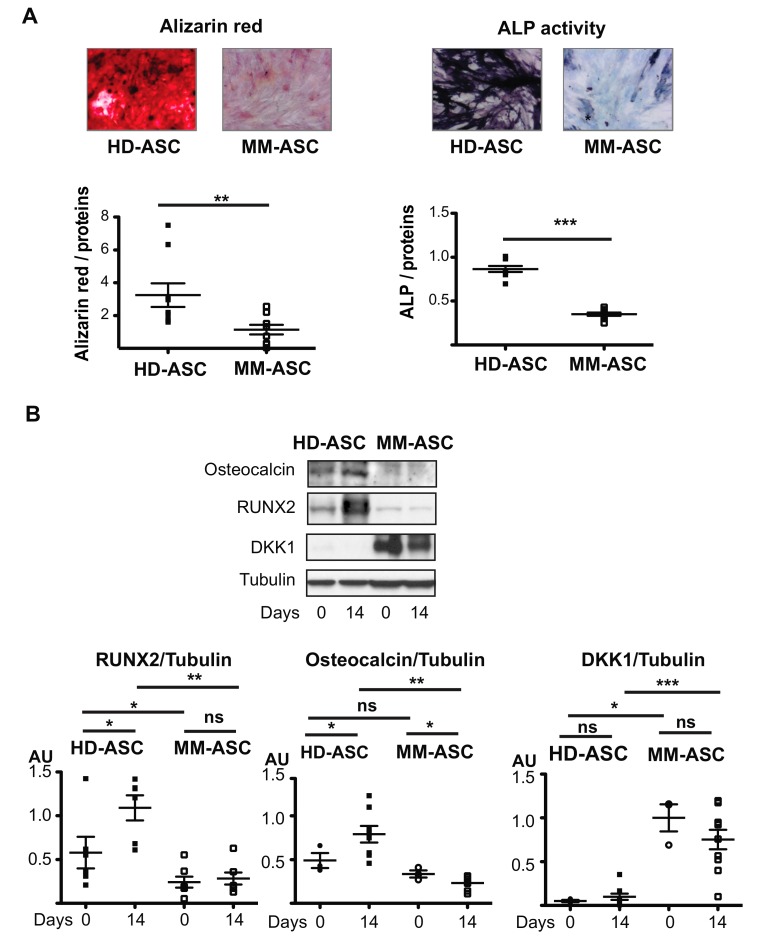
Osteoblast differentiation is altered in MM patients. ASC were differentiated into osteoblasts for 14 days. (**A**) The cells were stained with Alizarin Red to visualize calcium deposition and representative micrographs and scans are shown (left). Staining was quantified at 560 nm and normalized to the protein content. Alkaline phosphatase (ALP) activity was measured and representative micrographs and scans are shown (right). ** *p* < 0.01, *** *p* < 0.001, MM-ASCs vs. control cells (HD-ASC). (**B**) Whole cell lysates were extracted on day 0 or 14 of differentiation and analyzed by immunoblotting. Representative immunoblots of osteocalcin, RUNX2, DKK1, and tubulin (loading control) are shown in the upper panel and Western blot quantifications in the two lower panels. * *p* < 0.05, ** *p* < 0.01, *** *p* < 0.001. AU: Arbitrary units.

**Figure 4 cells-08-00441-f004:**
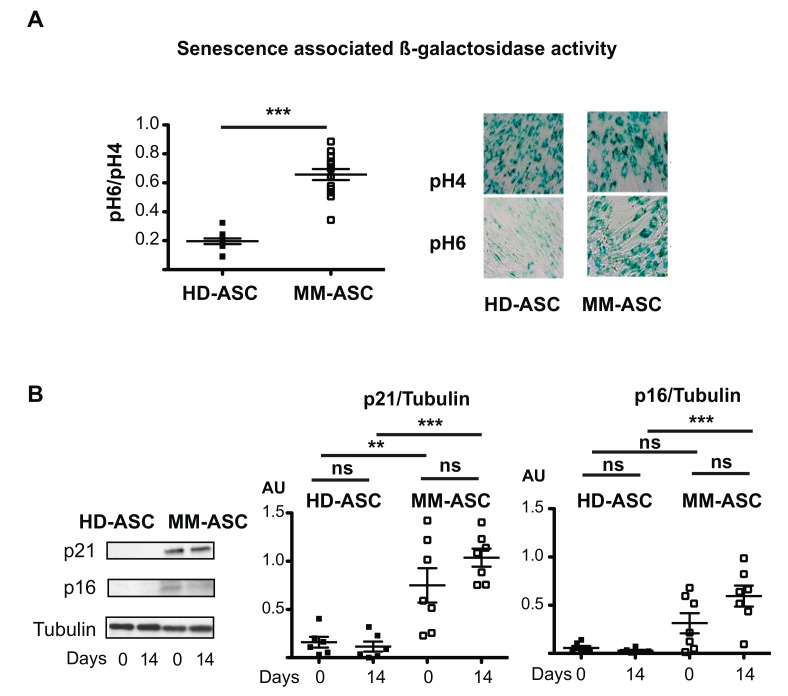
Senescence in MM-ASC versus HD-ASC. (**A**) SA β-galactosidase activity was assessed according to the ratio of pH 6- to pH 4-positive staining after 14 days of differentiation (left). Representative micrographs of β-galactosidase positive cells are shown (right). (**B**) Whole cell lysates were extracted from ASC on day 0 or 14 of differentiation and analyzed by immunoblotting. Representative immunoblots of the cell cycle arrest markers p21, p16, tubulin (loading control) are shown in the left panel and Western blot quantifications in the two right panels. * *p* < 0.05, ** *p* < 0.01, *** *p* < 0.001. AU: Arbitrary units.

**Figure 5 cells-08-00441-f005:**
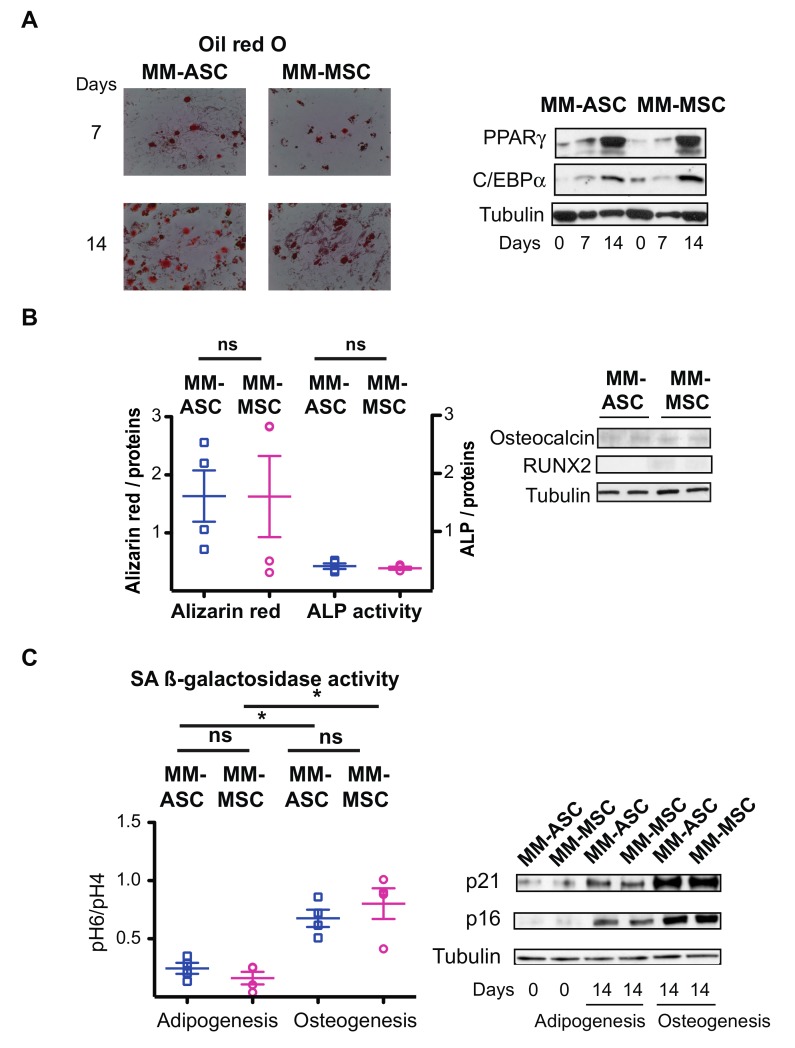
Comparison of ASC and MSC derived from the same MM patients. (**A**) MM-ASC and MM-MSC from the same patients were differentiated into adipocytes for 7 or 14 days. The cells were stained with Oil-red-O to visualize lipid droplets and representative micrographs and scans are shown (left). Whole cell lysates were extracted on day 0, 7, or 14 of differentiation and analyzed by immunoblotting. Representative immunoblots of PPARγ, C/EBPα, and tubulin (loading control) are shown (left). (**B**) MM-ASC (blue) and MM-MSC (pink) were differentiated into osteoblasts for 14 days. The cells were stained with Alizarin Red to visualize calcium deposition and ALP activity was assessed. ns, not significant (left). Whole cell lysates were extracted and analyzed by immunoblotting. Representative immunoblots of osteocalcin, RUNX2, and tubulin (loading control) are shown (right). (**C**) MM-ASC (blue) and MM-MSC (pink) were differentiated into adipocytes or osteoblasts for 14 days as indicated. SA-β-galactosidase activity was assessed according to the ratio of pH 6- to pH 4-positive staining after 14 days of differentiation. * *p* < 0.05, adipocyte vs. osteoblast. ns, not significant (left). Whole cell lysates were extracted from MM-ASC and MM-MSC on day 0 or 14 of adipocyte or osteoblast differentiation and analyzed by immunoblotting. Representative immunoblots of cell cycle inhibitors p21, p16, and tubulin (loading control) are shown (right).

**Figure 6 cells-08-00441-f006:**
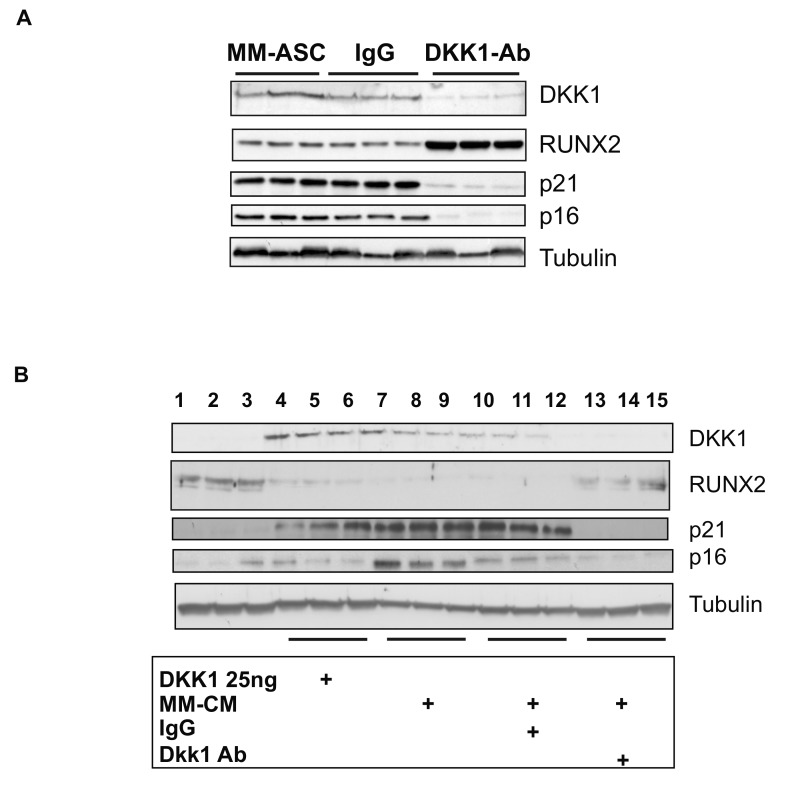
DKK1 inhibition rescues RUNX2 expression. (**A**) MM-ASC were cultured without or with an anti-DKK1 monoclonal antibody (DKK1-Ab) or IgG as control. Whole cell lysates were extracted and analyzed by immunoblotting. Representative immunoblots of RUNX2, DKK1, p16, p21, and tubulin (loading control) are shown. (**B**) The reverse experiment was performed by adding either recombinant DKK1 protein or conditioned medium (CM) from MM-ASC cultures to HD-ASC cultures, incubated or not with anti-DKK1. Immunoblots are shown.

**Table 1 cells-08-00441-t001:** Expression of hematopoietic markers in HD-ASC (*n* = 6) and MM-ASC (*n* = 11) at passage 2 (P2) of culture.

	HD-ASC	MM-ASC
CD45	<1%	<1%
CD14	<1%	<1%
HLA-DR	<1%	<1%
CD11b	<1%	<1%
CD19	<1%	<1%
CD34 (%)	5.6 ± 2.4	2.8 ± 0.9
CD34 MFI (AU)	2.1 ± 0.9	4.2 ± 2.05

Percentage of positive cells (mean ± SD) are indicated. Abbreviations: MFI: mean fluorescence intensity; AU: arbitrary units.

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
