# Peer review of "Systemic Dysfunction of Osteoblast Differentiation in Adipose-Derived Stem Cells from Patients with Multiple Myeloma"

_cells, 2019, doi:10.3390/cells8050441_

Round 1
Reviewer 1 Report
In this paper, the authors revealed that adipose-derived stem cells (ASC)isolated by fat aspiration from multiple myeloma patients (MM-ASC) have several biological alterations compared with ASC derived from healthy doners (HD-ASC) and certain roles in osteoblastogenesis in patients with MM. Especially it is a valuable evidence that DKK1 overexpression in MM-ASC derived from MM patients play important roles in bone disease of MM and suggests DKK1inhibition may be a novel therapy against lytic bone lesion. The bone pathology in multiple myeloma is very important in development of molecular target therapy against these bone diseases in order to improve the prognosis and quality of life of patients. However there are some defects in data and descriptions. Criticisms regarding this revised paper are discussed below.

Comments

1. There is no Figure 3. Legend only is showed in page 8.
2. Patients with MM are 10 persons including 3 patients without bone disease. Was there difference in osteoblast differentiation or senescence between 7 patients with bone disease and 3 patients without patients ? Please show the data of 3 patients without bone disease as other square or marking etc. in Figure 2, 3 and 4.
3. Figure 6 is very impressive data because it suggested that these data revealed cellular / molecular mechanisms of altered osteoblast differentiation / senescence in ASC in MM patients. So the readers may ask for the data about DKK1 expression in adipose tissue of MM patients and HDs using immunohistochemistry. Please show in situ expression of DKK1 in adipose tissues of MM patients and HDs.
Author Response
REVIEWER 1 COMMENTS TO AUTHORS:
In this paper, the authors revealed that adipose-derived stem cells (ASC)isolated by fat aspiration from multiple myeloma patients (MM-ASC) have several biological alterations compared with ASC derived from healthy doners (HD-ASC) and certain roles in osteoblastogenesis in patients with MM. Especially it is a valuable evidence that DKK1 overexpression in MM-ASC derived from MM patients play important roles in bone disease of MM and suggests DKK1inhibition may be a novel therapy against lytic bone lesion. The bone pathology in multiple myeloma is very important in development of molecular target therapy against these bone diseases in order to improve the prognosis and quality of life of patients. However there are some defects in data and descriptions.
Criticisms regarding this revised paper are discussed below.
1. There is no Figure 3. Legend only is showed in page 8.
Response: We apologize for this unfortunate oversight. The figure has been added page 8.
2. Patients with MM are 10 persons including 3 patients without bone disease. Was there difference in osteoblast differentiation or senescence between 7 patients with bone disease and 3 patients without patients? Please show the data of 3 patients without bone disease as other square or marking etc. in Figure 2, 3 and 4.
Response: To answer this interesting question regarding the correlation between the bone disease and the defect in osteogenic capacities, we have further analyzed our data. As now indicated in the supplemental Figure 1, patients with no bone lesion, at least at diagnosis and with the current imagine technics available, still have a similar defect of osteoblast differentiation together with an increased senescence (red triangles) as compared to patients with bone lesions. These data suggest that dysfunction of ASC is observed in MM patients even without bone disease.
The text has been changed accordingly. We added this analysis in the results section (page 7, paragraph 1: line 231-226; page 8, paragraph 1: line 275-279; page 9, paragraph 1: line 296-297) and we added a sentence in the discussion section (page 13, line 378-381).
3. Figure 6 is very impressive data because it suggested that these data revealed cellular / molecular mechanisms of altered osteoblast differentiation / senescence in ASC in MM patients. So the readers may ask for the data about DKK1 expression in adipose tissue of MM patients and HDs using immunohistochemistry. Please show in situ expression of DKK1 in adipose tissues of MM patients and HDs.
Response: We thank the reviewer for these positive comments. DKK1 immunostaining in adipose tissue from MM patients would indeed be a real argument in favor of our hypothesis.
However, as indicated in the Materials and Methods section, adipose tissues from MM patients were obtained by abdominal fat aspiration which is not compatible with adipose tissue architecture preservation. It is therefore not possible to perform immunochemistry with such samples. This point has been emphasized in the discussion section (page 14, line 407-409). Besides DKK1, other cytokines may have important role in the lytic bone lesions. We are currently analyzing a broad range of other potential cytokines.
In order to facilitate the understanding of the figure, the western blot quantifications are now in the Supplemental Figure 3.
Reviewer 2 Report
Bereziat et al. investigated the capability of adipose derived mesenchymal stem cells to differentiate into osteoblasts. The authors compared adipose derived MSCs from myeloma patients to healthy donors. They describe similarities but also differences between these 2 groups. Interestingly, the differentiation to functional osteoblasts was impaired for MScs derived from myeloma patients.
Major comments
What was the aim of this study? A clear hypothesis is missing. This could be better explained in the abstract and in the introduction. But be careful, the introduction is already lengthy and I would recommend shortening.
A paragraph on the limitations of the study is missing:
The number of patients is relatively low (N=11). The patients are poorly characterized. It is not clear whether the authors have investigated newly diagnosed patients or relapsed/pretreated patients. The healthy donors have not been characterized (age etc?).
In the case of pretreated patients, the authors cannot exclude effects of chemotherapy on MSCs. Another group of patients being treated for other malignant diseases then myeloma could serve as an appropriate control group.
It is barely appropriate to cultivate cells in vitro for several passages and then to perform functional assays. The gene expression profile will be considerably different after in vitro cultivation. This limitation has to be discussed.
Figure 3 is missing, only the legend was provided.
Author Response
REVIEWER 2 COMMENTS TO AUTHORS:
Bereziat et al. investigated the capability of adipose derived mesenchymal stem cells to differentiate into osteoblasts. The authors compared adipose derived MSCs from myeloma patients to healthy donors. They describe similarities but also differences between these 2 groups. Interestingly, the differentiation to functional osteoblasts was impaired for MScs derived from myeloma patients.
We would like to thank the Reviewer for the careful reading of the manuscript. We have addressed all the comments and changed the manuscript accordingly.
Major Comments and Suggestions for Authors
What was the aim of this study? A clear hypothesis is missing. This could be better explained in the abstract and in the introduction. But be careful, the introduction is already lengthy and I would recommend shortening.
Response: As requested by the reviewer, we have introduced a clear hypothesis and the aim of this study both in the abstract (page 1, line 34-39) and the introduction (page 2, line 80-92). The introduction has been also shortened.
A paragraph on the limitations of the study is missing:
Response: We have completed the discussion section with a paragraph concerning the limitations of our study (page 14, line 437-451).
The number of patients is relatively low (N=11). The patients are poorly characterized. The healthy donors have not been characterized (age etc?: )
Response: The characteristics of the patients are indicated in the supplemental table 1. We added the characterization of the control group (age, sex and BMI) in the materials and methods section (page 3, line 104-106).
4. It is not clear whether the authors have investigated newly diagnosed patients or relapsed/pretreated patients In the case of pretreated patients, the authors cannot exclude effects of chemotherapy on MSCs. Another group of patients being treated for other malignant diseases then myeloma could serve as an appropriate control group.
Response: As indicated in the section material and methods section, all the studied patients were newly diagnosed. There were no pretreated patients in this study.
It is barely appropriate to cultivate cells in vitro for several passages and then to perform functional assays. The gene expression profile will be considerably different after in vitro cultivation. This limitation has to be discussed.
Response: We agree with the reviewer and this limitation is now discussed (Page 14 line 437-451).
Figure 3 is missing, only the legend was provided.
Response: We apologize for this unfortunate oversight. The figure has been added Page 8.
Round 2
Reviewer 1 Report
In this paper, the authors revealed that adipose-derived stem cells (ASC)isolated by fat aspiration from multiple myeloma patients (MM-ASC) have several biological alterations compared with ASC derived from healthy doners (HD-ASC) and certain roles in osteoblastogenesis in patients with MM. Especially it is a valuable evidence that DKK1 overexpression in MM-ASC derived from MM patients play important roles in bone disease of MM and suggests DKK1inhibition may be a novel therapy against lytic bone lesion. The bone pathology in multiple myeloma is very important in development of molecular target therapy against these bone diseases in order to improve the prognosis and quality of life of patients. The authors revised this paper at manner of point-by-point.

Comments

1. Figure 3 was added.
2. The supplement figures show each data of patients with MM are 10 persons including 3 patients without bone disease.
3. I expect that the data about DKK1 expression in adipose tissue of MM patients and HDs using immunohistochemistry are shown in next paper.
Reviewer 2 Report
The authors have addressed my criticism.